# Decoupling Pixel Flipping and Occlusion Strategy for Consistent XAI Benchmarks

**Stefan Blücher**                                   *bluecher@tu-berlin.de*
*BIFOLD – Berlin Institute for the Foundations of Learning and Data*
*Machine Learning Group, TU Berlin*

**Johanna Vielhaben**                          *johanna.vielhaben@hhi.fraunhofer.de*
*Explainable Artificial Intelligence Group*
*Fraunhofer Heinrich-Hertz-Institute*

**Nils Strodthoff**                                  *nils.strodthoff@uol.de*
*Division AI4Health*
*Carl von Ossietzky Universität Oldenburg*

**Reviewed on OpenReview:** *https://openreview.net/forum?id=bIiLXdtUVM*

## Abstract

Feature removal is a central building block for eXplainable AI (XAI), both for occlusion-based explanations (Shapley values) as well as their evaluation (pixel flipping, PF). However, occlusion strategies can vary significantly from simple mean replacement up to inpainting with state-of-the-art diffusion models. This ambiguity limits the usefulness of occlusion-based approaches. For example, PF benchmarks lead to contradicting rankings. This is amplified by competing PF measures: Features are either removed starting with most influential first (MIF) or least influential first (LIF).

This study proposes two complementary perspectives to resolve this disagreement problem. Firstly, we address the common criticism of occlusion-based XAI, that artificial samples lead to unreliable model evaluations. We propose to measure the reliability by the R(eference)-Out-of-Model-Scope (OMS) score. The R-OMS score enables a systematic comparison of occlusion strategies and resolves the disagreement problem by grouping consistent PF rankings. Secondly, we show that the insightfulness of MIF and LIF is conversely dependent on the R-OMS score. To leverage this, we combine the MIF and LIF measures into the symmetric relevance gain (SRG) measure. This breaks the inherent connection to the underlying occlusion strategy and leads to consistent rankings. This resolves the disagreement problem of PF benchmarks, which we verify for a set of 40 different occlusion strategies.

## 1 Introduction

Explainable AI (XAI) reveals the reasoning structure of black-box machine learning (ML) models (Lundberg & Lee, 2017; Montavon et al., 2018; Samek et al., 2019; Covert et al., 2021; Samek et al., 2021). Appropriate usage of XAI methods enables new research avenues in various scientific domains (Holzinger et al., 2019; Blücher et al., 2020; Binder et al., 2021; Anders et al., 2022; Klauschen et al., 2024). However, a multitude of possible XAI methods leads to practical challenges (Freiesleben & König, 2023). For example the inconclusive scenario of multiple, contradictory explanations for a single model prediction was recently dubbed as disagreement problem (Neely et al., 2021; Krishna et al., 2022). However, such a disagreement problem not only arises in explanations itself, but extends to their evaluation. In pixel flipping (PF) (Samek et al., 2016), which assesses the faithfulness of XAI methods by removing features from the model prediction depending on their explanations, the final ranking of XAI methods depends on the specific PF setup (see Table 1). On one side, either removing most-influential features (MIF) or least-influential features (LIF) first when performing

Table 1: *Which ranking do you pick?* The choice of measure (most influential first (MIF) vs. least influential first (LIF)) and occlusion strategy (train set vs. diffusion) influences the ranking of XAI methods. Saliency, layer-wise relevance propagation (LRP) and integrated gradients (IG) denote three widely used attribution methods.

| | **Disagreement problem of PF** | | | |
|---|---|---|---|---|
| Setup | MIF | LIF | MIF | LIF |
| | Train set | | Diffusion | |
| Ranking | IG | LRP | LRP | IG |
| | LRP | Saliency | Saliency | LRP |
| | Saliency | IG | IG | Saliency |

PF benchmarks leads to disagreeing rankings of XAI methods. On the other side, the underlying occlusion strategy can be implemented in various different ways. This leads to further disagreeing PF rankings and also to multiple (contradictory) occlusion-based explanations (Covert et al., 2021).

This study provides a two-fold contribution: Firstly, we address the main criticism of occlusion-based XAI approaches, which states that occluded samples are artificial and thus their evaluation is potentially not reliable (Gomez et al., 2022). We quantify this concern via the Reference-out-of-model-scope (R-OMS) score and thereby enable an objective comparison of occlusion strategies. Secondly, we thoroughly analyze the disagreement problem of PF benchmarks. Here, sorting PF setups based on the R-OMS score groups consistent rankings. Moreover, MIF and LIF ranking are complementary, in the sense that rankings are consistent for either measure across the R-OMS spectrum. Based on this intuition, we propose the symmetric relevance gain (SRG), which combines both measures. The SRG measure provides consistent rankings across all occlusion strategies and thereby resolves the disagreement problem of PF benchmarks.

## 2 Crucial role of occlusion strategies for XAI

Before investigating occlusion strategies in detail, we first discuss their usage in the context of both XAI methods and their evaluation. We introduce our notation in Table 2.

Table 2: Notation.

| | | | |
|---|---|---|---|
| $N = \{1, 2, \ldots, n\}$ Feature set | | $S \subseteq N$ Coalition | $s = |S|$ Cardinality |
| $x = (x_1, \ldots, x_n)$ | Specific sample: $x_i$ might be aggregated input features (superpixels) | | |
| $X = (X_S, X_{\bar{S}})$ | Generic (random) sample spit into complements $\bar{S} = N \backslash S$ | | |
| $\pi = [\pi_1, \ldots, \pi_n]$ | Feature ordering | $\Pi(s) = \{\pi_i | i \leq s\}$ | Leading features in $\pi$ |
| $\pi^r = [\pi_n, \ldots, \pi_1]$ | Reverse feature ordering | | |
| $f_c(x)$ Classification model $p(c|x) \in [0, 1]$ | | $\mathfrak{f}_c(x_S)$ | Occluded model restricted to $S$ |

### 2.1 Understanding model reasoning with XAI

Generally, XAI research is concerned with verifying and revealing the reasoning structure of machine learning models (Lapuschkin et al., 2019). Various approaches have been proposed such as attribution methods (Baehrens et al., 2010; Bach et al., 2015; Ribeiro et al., 2016; Zintgraf et al., 2017; Letzgus et al., 2022), concept discovery (Kim et al., 2018; Ghorbani et al., 2019; Vielhaben et al., 2023; Achtibat et al., 2023) or global (model-wide) analysis tools (PDP) (Hastie et al., 2009)/ALE (Apley & Zhu, 2020)/ICE (Goldstein et al., 2015). Here, we focus on model-agnostic approaches, which all build on occluding features and observing changes in the model prediction (Covert et al., 2021).

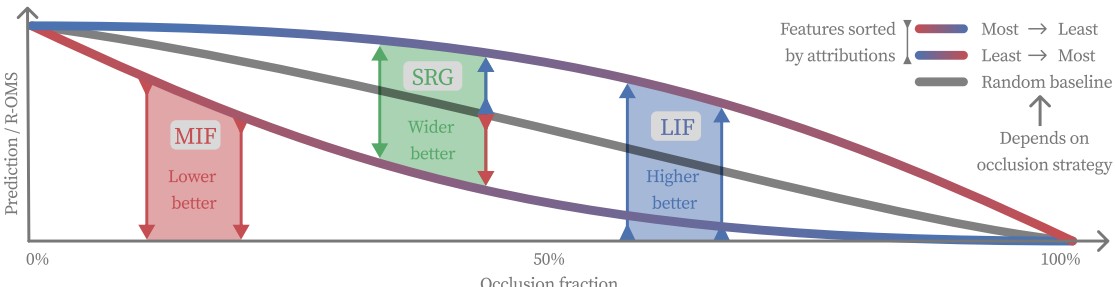

Figure 1: Pixel flipping benchmarks of XAI methods. Both MIF and LIF are affected by the random baseline. Using the complete symmetric relevance gain (SRG) introduced in Section 5.2 breaks the inherent dependence on the occlusion strategy.

**Model-agnostic XAI methods rely on occluding features** Here, Shapley values are a widespread approach. They build on an abstract value function $v : 2^n \to \mathbb{R}$, which maps a feature set to a scalar payout. The attribution of feature $i$ is given by its average marginal gain:

$$\phi_i = \sum_{S \subseteq N \setminus i} \mathcal{N}(s) \left[ v(S \cup i) - v(S) \right] . \tag{1}$$

The unique normalization $\mathcal{N}(s) = \frac{s!(n-s-1)!}{n!}$ ensures the Shapley axioms symmetry, linearity, efficiency and null player (Shapley, 1953). To deal with the binomial growth in coalitions $S$, we approximate Equation (1) by uniformly sub-sampling coalitions (Štrumbelj & Kononenko, 2010; Lundberg & Lee, 2017). Competing XAI methods such as PredDiff (Robnik-Sikonja & Kononenko, 2008; Zeiler & Fergus, 2014; Zintgraf et al., 2017; Blücher et al., 2022) only remove the target feature or introduce the target feature into a fully occluded sample such as ArchAttribute (Tsang et al., 2020). PredDiff and ArchAttribute correspond to the first and last marginal contribution in Equation (1) respectively. The last step is to connect the abstract value function with the occluded model prediction via $v(S) = \log \mathfrak{f}_c(x_S)$ (Blücher et al., 2022). Therefore, ambiguities related to the occlusion strategy (called design choices in Section 3) are contained within the resulting XAI method. This leads to multiple Shapley values despite the underlying uniqueness property (Sundararajan & Najmi, 2020).

## 2.2 Evaluation of XAI methods

**Pixel flipping for attribution evaluation** To systematically judge the quality of XAI methods, a variety of approaches has been proposed (Nauta et al., 2023). This ranges from pixel flipping (Samek et al., 2016; Rieger & Hansen, 2020; Samek et al., 2021; Gevaert et al., 2022; Hedström et al., 2023; Li et al., 2023) over sanity checks (Adebayo et al., 2018; Binder et al., 2023) to synthetic datasets with ground truth knowledge (Yang & Kim, 2019; Kayser et al., 2021; Budding et al., 2021; Arras et al., 2022).

Here, we focus on pixel flipping, as a general and widespread solution for quantitative evaluation of XAI methods (Samek et al., 2016). PF measures the faithfulness of attributions: Is the actual model behavior captured by the attributions? To this end, one summarizes the changes in model prediction after successively removing features depending on some ordering $\pi$:

$$\text{AUC}[\pi] = \frac{1}{n} \sum_{s=0}^{n} v^{\text{PF}}(\Pi(s)). \tag{2}$$

In analogy to Shapley values, the value function $v^{\text{PF}}(S) = \mathfrak{f}_{\hat{c}}(x_S)$ denotes the occluded model prediction. A given explanation $\phi$ induces a unique feature ordering $\pi^{\phi} = \arg\text{sort}_i \phi_i$. Then the faithfulness of the underlying XAI method is assessed by the PF measures

(Samek et al., 2016; Petsiuk et al., 2018; Samek et al., 2021; Gomez et al., 2022; Brocki & Chung, 2023):

$$\text{MIF}[\phi] = \text{AUC}\left[\pi^\phi\right] \qquad \text{(higher better)}$$
$$\text{LIF}[\phi] = \text{AUC}\left[\left(\pi^\phi\right)^r\right] \qquad \text{(lower better)}$$

(3)

Faithful attributions lead to a steep (most relevant first, MIF) or flat (least relevant first, LIF) descent for the two opposing measures (colored curves in Figure 1). Complementary literature proposed to occlude a fixed number of features (sensitivity-$n$) and measure the calibration (sum of attributions) compared to the actual drop in model prediction. (Ancona et al., 2017; Yeh et al., 2019; Bhatt et al., 2020).

**PF rankings rely on occluding features** Like all model-agnostic XAI methods, PF benchmarks share all design choices of the underlying occlusion strategy (Tomsett et al., 2020). Thus, PF setups are ambiguous (Gevaert et al., 2022; Rong et al., 2022; Barr et al., 2023) and lead to disagreeing rankings (Table 1). Unfortunately, PF is commonly invoked to *demonstrate* the superiority of a newly proposed XAI method (Freiesleben & König, 2023). For practical reasons, studies naturally focus on a single PF setup and neglect the inherent variability. Therefore, this study investigates influential factors of occlusion strategies that can effect the final method ranking for MIF/LIF benchmarks.

## 2.3 Occlusion strategies in the literature

Previous research has investigated different possibilities for occluding features. Here, baselines that mimic feature absence by a constant value are a widely used option. The impact of the specific value has been investigated and visualized in various studies (Sturmfels et al., 2020; Haug et al., 2021; Mamalakis et al., 2022). Complementary, Izzo et al. (2020); Shi et al. (2022); Ren et al. (2023) proposed criteria to fix the baseline value. However, Jain et al. (2022) showed that simple baselines can lead to undesirable out-of-model-scope biases. Exposing the model to artificially occluded samples during training can circumvent this problem (Hooker et al., 2019; Hase et al., 2021; Brocki & Chung, 2023). Alternatively, improved occlusion strategies can be employed to create realistic in-distribution samples (Kim et al., 2018; Chang et al., 2019; Agarwal & Nguyen, 2020; Sivill & Flach, 2022; Olsen et al., 2022; Rong et al., 2022; Augustin et al., 2023) Lastly, XAI methods can be adjusted to compensate for OMS effects (Dombrowski et al., 2019; 2022; Qiu et al., 2021; Fel et al., 2023; Taufiq et al., 2023; Dombrowski et al., 2023) and prevent adversarial vulnerabilities (Anders et al., 2020; Slack et al., 2020). However, the central question of how to choose reliable occlusion strategies remains still unsolved.

## 3 Design choices for occlusion strategies

The occlusion strategy is commonly identified with the imputer distribution. This is insufficient as more design choices can impact the reliability of occluded samples such as size and shape of superpixels. In particular, occlusion strategies are inherently connected to the underlying model, which relies on both architecture and training procedure. This study addresses computer vision, but similar considerations apply to other domains.

### 3.1 Design choice I: Imputer

**Feature removal paradigms** Occluded model predictions are a ubiquitous component of XAI (see Section 2). Generally, it is not possible to *omit* features $\bar{S}$ for the occluded prediction $\mathfrak{f}_c(x_S)$, but one has to *shield* the model prediction from their impact. Here, the only model-agnostic option is to construct occluded samples $(x_S, X_{\bar{S}})$ based on an imputer $q$, which generates artificial values $X_{\bar{S}}$. Then the occluded model prediction is given by $\mathfrak{f}_c(x_S) = \sum_{X_{\bar{S}} \sim q} \mathfrak{f}_c(x_S, X_{\bar{S}})$. There are two principled possibilities for the imputer: Firstly, the **conditional** distribution $q = p(X_{\bar{S}}|x_S)$ allows to exactly marginalize the complementary features $\bar{S}$, i.e., $\mathfrak{f}_c(x_S) = p(c|x_S) = \int dX_{\bar{S}} p(c|x_S, X_{\bar{S}}) p(X_{\bar{S}}|x_S)$. This relation lies at the heart of PredDiff (Blücher et al., 2022). Secondly, the **marginal** distribution $q = p(X_{\bar{S}})$ explicitly breaks the relation between the features $S$ and $\bar{S}$. Due to this independence, marginal imputers are easily accessible and enable causal interpretations (Janizek et al., 2020). In our experiments, we consider the set of imputers listed in Table 3.

The alternative model-specific option is to leverage **internal** structures to remove features $\bar{S}$: Some models allow to directly omit features in a meaningful manner (from the perspective of the model). Here, examples are tree-based models (Lundberg et al., 2020) or transformers (Jain et al., 2022).

Table 3: Practical imputers considered in our experiments, ranging from simple to complex.

| Imputer | Example | Description |
|---|---|---|
| **Mean** *marginal deterministic* | | Occludes superpixels with constant channel-wise data set mean. |
| **Train set** *marginal probabilistic* | | The imputed features are drawn from a random training set sample (optimal marginal imputer). |
| **Histogram** *conditional probabilistic* | | (Wei et al., 2018) Occludes superpixels with constant value sampled from colors contained in image (conditional analogue of the mean imputer). |
| **cv2** *conditional deterministic* | | Inpaints occluded superpixels based on the surrounding pixel values (Telea, 2004). |
| **Diffusion** *conditional probabilistic* | | Inpaints using a (class-unconditional) state-of-the-art diffusion model. Imputations are visually more aligned since the remaining features are used as reference (Lugmayr et al., 2022). |

## 3.2 Design choice II: Superpixel shape and number

**From arbitrary features to superpixels**

We now consider the case of images $x \in \mathbb{R}^{w \times h \times n_{\text{channels}}}$ with width $w$, height $h$ and $n_{\text{channels}}$ color channels. Here, single pixels share redundant information with neighboring pixels. It is therefore more meaningful to consider a collection of pixels (superpixels) as individual features. Superpixels are obtained by segmenting a complete image into disjoint patches

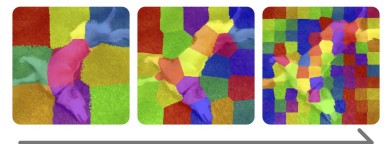

Number of superpixels $n$

$N = \{1, 2, \ldots, n\}$. Computational costs of the attribution method then depend on the number of superpixels $n \ll w \cdot h \cdot n_{\text{channels}}$. All considered segmentation algorithms are listed in Table 4.

## 3.3 Design choice III: Model

How occluded samples are perceived by the model under consideration, can depend on the model architecture or its training procedure. Therefore, we use three different models in this study: Firstly, we use the standard-ResNet50 (He et al., 2016) as provided by torchvision. Secondly, we compare to the same architecture but trained with a state-of-the-art training procedure (Wightman et al., 2021) as provided by the timm library (timm-ResNet50). Lastly, we investigate a vision transformer (ViT) model, which was already used in (Jain et al., 2022) to demonstrate the effects of occlusion strategies.

# 4 Comparing occlusion strategies via the R-OMS score

In this section, we derive a quantitative approach to characterize artificial samples solely relying on model predictions. This enables a systematic comparison of occlusion strategies - thereby judging the impact of the underlying design choices.

Table 4: Superpixel can be generated by simple or more advanced segmentation algorithms

**Rectangular patches**: Simple fixed segmentation mask, which is independent of the image.

**Classical segmentations**: Segmentation aligned to the semantic image content to some degree. Here, we use the classical SLIC algorithm (Achanta et al., 2012) with default compactness $\lambda = 1$.

**Semantic segmentation**: A meaningful, semantic segmentation for each image is on the advanced end of the spectrum, Here, we build on the Segment Anything Model (SAM) (Kirillov et al., 2023).

### 4.1 Quantitative strategy to assess occluded samples

**Artificial samples are not in-distribution** Occlusion-based XAI relies on imputed samples, which are necessarily artificial. This is a common point of criticism, as the model needs to some extent extrapolate away from the original data manifold (Hooker & Mentch, 2019; Kumar et al., 2020). This is closely related to the out-of-distribution detection community, which aims to detect unreliable predictions to enable monitoring and employment of ML for safety-critical applications (Salehi et al., 2022). Here, many studies strived to quantify whether a given sample is out-of-distribution with respect to the underlying data distribution. However, this perspective neglects the specific model under consideration. Therefore, a recent study suggested (Guérin et al., 2023) that it is more meaningful to characterize the out-of-model-scope-ness (OMS-ness) of samples. We also adopt this perspective here, as the model is the crucial component underlying any XAI application.

**Relying on the reference samples is essential** Conventional OMS scenarios monitor the reliability of arbitrary samples. This is in stark contrast to occlusion-based XAI, which is interested in the reliability of *artificial* imputations of a single fixed *original* sample. Therefore, the original sample serves as a reference and one is interested in the difference with respect the occluded sample. In analogy to the image quality assessment literature (Kamble & Bhurchandi, 2015), we refer to scores that leverage the available side-information about the original sample, as Reference-OMS (R-OMS). In contrast, conventional OMS-measures are denoted as No-Reference-OMS (NR-OMS) scores. Conceptually, this distinction allows for characterizing and potentially adapting any OMS measure.

**Original class prediction as R-OMS score** A simple NR-OMS-score is the maximum softmax probability (MSP) $\max_c f_c(x_S)$ (Hendrycks & Gimpel, 2016). For a low MSP score, the model is not confident about its prediction and thus the input sample is unreliable. To obtain the related R-OMS score, we focus on the original class prediction $f_{\hat{c}}(x_S)$ by leveraging the original sample $x$ and its label $\hat{c}$. From here on, R-OMS-score refers to this measure. The R-OMS score tracks how much information about the original sample is still accessible to the model. A high R-OMS score is indicative of reliably occluded samples.

**Comparison of R-OMS and NR-OMS**
For occluded samples, the R-OMS and NR-OMS score agree as long as there is no flip in the predicted class label. This changes for severely altered imputations. We exemplified this in two distinct cases. Firstly, for samples that are imputed with some natural image content (train set or diffusion), the generated features inevitably correspond to some other output class ($\uparrow$ NR-OMS). Thus, the occluded sample is biased towards the class that the train set imputer draws or the diffusion imputer inpaints. A possible workaround might be to average over multiple samples when calculating the occluded prediction. However, this introduces linearly increasing computation costs. Secondly, the missingness bias (Jain et al., 2022) trig-

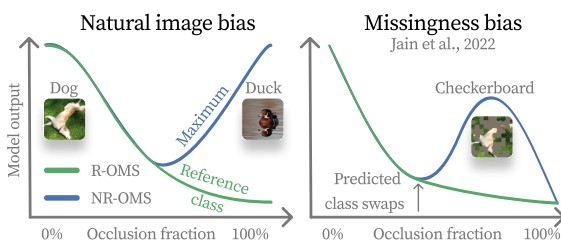

Figure 2: R-OMS vs. NR-OMS.

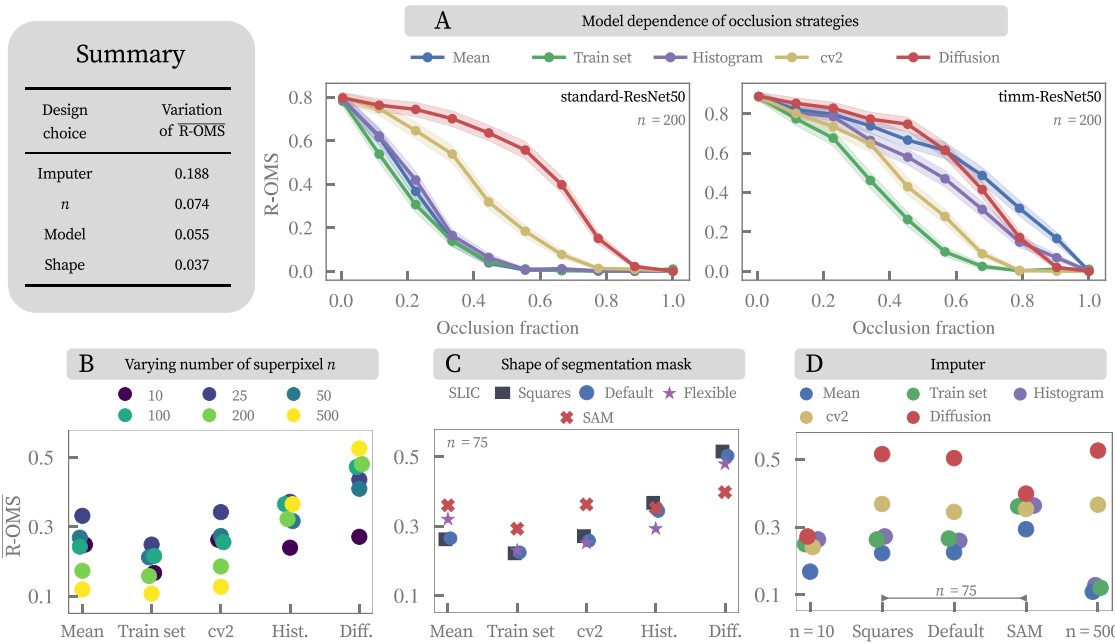

Figure 3: (A): Occlusion strategy and model interact. (B, C, D): Visualize variation of each design choice for fixed standard-ResNet50. (Summary) reports the average variation (interquartile ranges) associated with each design choice. (B) Granularity of segmentation (C) Shape of segmentation. (D) Imputer choice.

gers specific classes (e.g. checkerboard) based on the occlusion strategy (↑ NR-OMS). The R-OMS is not affected by both effects as these are unrelated to the reference class (↓ R-OMS). Therefore the R-OMS-score typically decreases monotonically whereas the NR-OMS-score potentially rises again when occluding more patches.

## 4.2 Characterize the impact of design choices

**R-OMS coincides with the PF target** In the following, we characterize occlusion strategies by measuring the R-OMS score at various occlusion fractions $(n - s)/n$ for a random set of occluded superpixels $S$. A steeper descent reflects less reliable occlusion strategies. Importantly, this approach is independent of the considered XAI method. The AUC of this random PF curve is identical to the random baseline in PF benchmarks. To stress this inherent connection, this value is denoted as $\overline{\text{R-OMS}} = \mathbb{E}_{\text{uni}(\pi)}[\text{AUC}[\pi]]$.

**Occlusion strategies depend on the model choice** In Figure 3 (A), we compare the standard ResNet50 vs. timm-ResNet50. Even though both models rely on the same architecture, the R-OMS scores for identical imputers (mean and histogram) vary significantly (Crothers et al. (2023) observed this in the NLP context). Occluded samples with mean-imputed superpixels are not similar to natural images (see example in Table 3). Therefore, one naively expects a low R-OMS score as it occurs for the standard ResNet50. However, the timm-ResNet50 confidently predicts the correct class even for heavily occluded samples. This change in model response originates from the different training schemes. Timm-training invokes an elaborate augmentation procedure that boosts model performance (Wightman et al., 2021). Thereby, the timm-ResNet50 learned to ignore non-informative constant patches and solely focus on the remaining original image content. Importantly, the R-OMS detects unreliable samples, as judged by the trained model, without knowledge about the original training strategy. Alternatively, one can enforce reliability by manually pre-training models with occluded samples, however at the price of a large computational overhead (Hooker et al., 2019; Hase et al., 2021). The above observation exemplifies, that human judgment of occluded samples is inherently flawed, as it lacks any connection to the underlying model.

**Diffusion ensures reliable model predictions** The subfigures (A) and (D) in Figure 3 show that the diffusion imputer consistently leads to the highest R-OMS scores. This is expected, as generative diffusion models are trained to inpaint realistic patches for the masked superpixels. Therefore, occluded samples are similar to images seen during training for both models. This is in contrast to the marginal train set imputer, which leads to unrealistic samples (low R-OMS). Interestingly, the imputed superpixels themselves are drawn from the original data manifold and are therefore natural to the model. However, the model is confused by the contradictory (imputed vs. original) information. As a consequence, the model cannot leverage the remaining information from the original image. Thus, the train set imputer, which is the optimal marginal approach, leads to a steep descent in model confidence.

**Diffusion imputer resemble internal strategy** To further characterize the diffusion imputer, we compare it to the internal occlusion strategies of a ViT model. Here, we summarize our findings and defer the exact results to the supplementary Figure A1. The internal imputer is a neutral approach (not relying on artificial samples), as the masked superpixels are directly omitted (Jain et al., 2022). Based on the R-OMS score, we find a close alignment between the internal and diffusion strategy. This is very unexpected since both occlusion mechanisms are conceptually very different. To further strengthen this point, we confirmed that this similarity extends to the intermediate hidden activations. Overall, this alignment is an interesting argument in favor of the diffusion imputer as a natural replacement strategy.

**Number of superpixels significantly impacts all imputers** We compare the occlusion strategies depending on the number of superpixels in Figure 3 (C). We observe that simple imputers (mean, histogram and train set) are more reliable for larger superpixels (small $n$). Contrary, for smaller superpixels, a missingness bias (see also Figure 2) reduces the $\overline{\text{R-OMS}}$ score. An opposing trend is visible for the conditional imputers (cv2 and diffusion), for which artificial samples are increasingly realistic to the model with smaller superpixels. This is an expected behavior, since for fixed occlusion fraction the imputation task for a smaller number of superpixels is comparably simpler than the same task for a larger number of superpixels, where larger segments have to be inpainted in a semantically meaningful fashion.

**Naive imputations are more meaningful for semantic superpixels** Next, we investigate how the segmentation algorithm (shape of superpixels) impacts the occlusion strategies (C). As outlined in Table 4, we compare squares, default-SLIC ($\lambda = 1$), flexible-SLIC ($\lambda = 0.1$) and semantic SAM superpixels (Yeh et al., 2019; Yu et al., 2023). To ensure a fair comparison we filter for images with a similar number of superpixels $n \approx 75$. We observe that semantic SAM superpixels increase the $\overline{\text{R-OMS}}$ score for all three simple imputers. This aligns with (Rong et al., 2022), who discussed a similar phenomenon as information leakage through the segmentation mask, an effect which could also be framed as a positive missingness bias (Jain et al., 2022). In contrast, the conditional cv2 and diffusion imputer struggle to meaningfully embed semantic patches into the local neighborhood. Thus, the $\overline{\text{R-OMS}}$ score decreases for increasingly flexible superpixels. Overall, semantic superpixels reduce the influence of the imputer choice as apparent from (D). Here, SAM superpixels clearly show the least variation.

**Relative importance of design choices** So far, we discussed each design choice individually. In particular, we aimed to vary each dimension between its two extremes (small vs. large superpixels, square vs. semantic superpixel shapes, simple vs. complex imputers). Each design choice therefore induces an inherent degree of variation into the resulting occlusion strategy. To quantitatively assess this variation, we calculate the maximal $\overline{\text{R-OMS}}$ spread for each design choice (columns in B, C and D) and report the interquartile range (ICR) over all experiments in the summary table in Figure 3. The supplementary Table A1 shows details about the underlying setups for the model. Based on this analysis, we conclude that the imputer choice is the dominant design choice of the occlusion strategy. The secondary effects are associated with the number of superpixels and the underlying model. Lastly, the shape of superpixels has the least impact on the occlusion strategy. As a consequence, it is generally not worth invoking expensive SAM superpixels to obtain more reliable occlusion strategies. In fact, this can even be detrimental when employed in conjunction with the diffusion imputer.

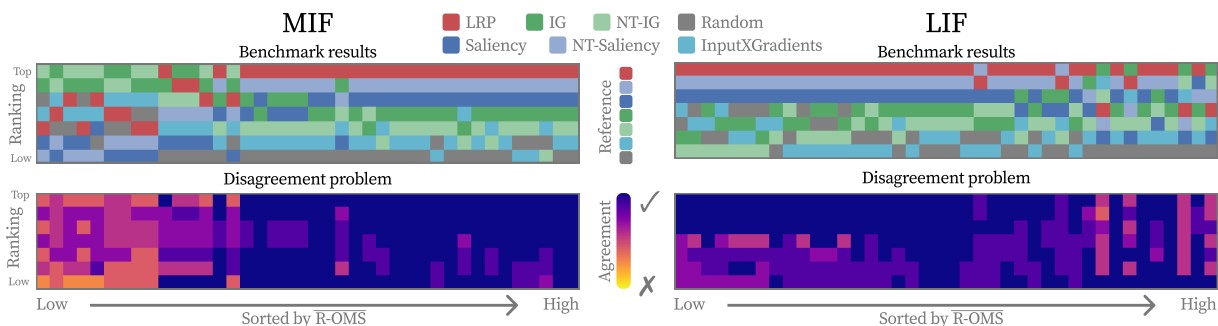

Figure 4: PF benchmarks based on varying occlusion strategies lead to many disagreeing rankings for both MIF and LIF. Sorting rankings based on the $\overline{\text{R-OMS}}$ groups consistent rankings. The lower panel visualizes the disagreement problem as the deviation from the most frequent ranking (reference). The consistency of MIF (high) and LIF (low to medium) are complementary when sorting based on the $\overline{\text{R-OMS}}$.

## 5 Impact of occlusion strategies on PF benchmark

**Setup** This section explores the impact of different occlusion strategies on PF benchmarks. All results are based on 100 randomly selected imagenet samples. Based on Section 3, we construct a diverse set of 40 occlusion strategies, varying all design choices ($n$: 25, 100, 500, 5000; imputer: mean, train set, histogram, cv2, diffusion; model: standard-ResNet50, timm-ResNet50). Using all 40 PF setups we rank several standard XAI methods: Saliency, (NT)-Saliency with Noise-tunnel, Integrated gradients (IG), InputXGradients, layer-wise relevance propagation (LRP) and the random baseline ($\overline{\text{R-OMS}}$). Pixel-wise attributions are averaged within a superpixel to obtain superpixel-based attributions. Gradient-based attributions are calculated using captum (Kokhlikyan et al., 2020), LRP using zennit (Anders et al., 2021) and model-agnostic attributions based on a custom implementation. Our code is available on at https://github.com/bluecher31/pixel-flipping.

### 5.1 Disagreement problem of MIF and LIF

**Occlusion strategies lead to many rankings** Based on all 40 PF setups we perform both MIF and LIF benchmarks. Varying the occlusion strategy leads to 17 (MIF) and 23 (LIF) different rankings (top panel in Figure 4). At this stage, it is not obvious which ranking is the most trustworthy. In principle, an adversary can advocate any method, by selecting the PF setup for which the method performs best. This is very troublesome and prevents a fair comparison of XAI methods in terms of faithfulness. This exemplifies the prevalence of the disagreement problem for PF benchmarks (lower panel).

**Identifying consistent MIF and LIF rankings** Next, we analyze the collection of all rankings in detail. To this end, we sort the PF setups based on the underlying $\overline{\text{R-OMS}}$ score and visualize all rankings accordingly in Figure 4. Interestingly, the rank of the random baseline seems to be also sorted by this. For low $\overline{\text{R-OMS}}$ the random baseline consistently outperforms established methods for both LIF as well as MIF. This verifies that occlusion strategies with low $\overline{\text{R-OMS}}$ score are indeed not reliable. For the top-ranked methods LIF and MIF behave conversely. This phenomenon is rooted in the opposing orientation of the insightfulness of both measures, which we discuss in next section Section 5.2. MIF rankings are most consistent for a high $\overline{\text{R-OMS}}$ whereas the top-ranking LIF methods agree for a low score. The lower panel in Figure 4 visualizes this consistency based on the deviation to the most frequent ranking, which is identical for MIF and LIF. Clearly, MIF rankings are fully consistent for large $\overline{\text{R-OMS}}$. For the LIF measure the situation is not as conclusive. Here, a medium score seems to lead to the most consistent rankings. Overall, the $\overline{\text{R-OMS}}$ can be viewed as an observable, which characterizes the outcome of PF benchmarks.

**Quantitative characterization of PF benchmarks** Our qualitative analysis showed, that the $\overline{\text{R-OMS}}$ score groups consistent rankings for both MIF and LIF benchmarks. To quantify this notion, we define a (ground truth) consistency score for all rankings as the normalized discounted cumulative gain (nDCG)

Table 5: Sorting rankings based on different variables. A high score means that a variable groups similar rankings (high consistency) and therefore characterizes the PF setup. Ground truth sorting is defined based on the nDCG. Variance $\sigma$ indicates the consistency of randomly sorting rankings (zero on average $\mu = 0$).

| Design choices | MIF | LIF | Observables | MIF | LIF |
|---:|:---:|:---:|:---:|:---:|:---:|
| # superpixels $n$ | 0.57 | 0.53 | $\overline{\text{R-OMS}}$ | 0.79 | 0.45 |
| Imputer | 0.57 | 0.41 | $\overline{\text{NR-OMS}}$ | 0.59 | 0.11 |
| Model | 0.23 | 0.22 | Random $[\pm\sigma]$ | 0.16 | 0.16 |

(Järvelin & Kekäläinen, 2002) with respect to the most frequent ranking. The nDCG penalizes misses in the leading position (winning XAI methods) more severely as changes at later position (around the random ranking). In Table 5 we report the correlation between the nDCG score and variables associated with the PF setup. Variables with a high correlation are predictive of the resulting ranking of the PF benchmark. The number of superpixels $n$ is the most indicative design choice. However, design choices do not provide an objective criterion to distinguish PF setups. For example, consider the imputer choice, which does not have an inherent ordering. To circumvent this, we probed for all possible imputer orderings and reported the maximum correlation. In contrast, the (N)R-OMS scores are observables that naturally order PF setups. From Table 5 it is clear that using the reference sample (R-OMS) to characterize the occlusion strategy is beneficial and leads to a more consistent sorting. Using a naive non-reference measure (NR-OMS) is less insightful and even leads to a nearly random performance for the LIF benchmark.

## 5.2 Consistent PF benchmarks with the SRG measure

**Insightful MIF/LIF setups depend on the R-OMS score** We just saw that the $\overline{\text{R-OMS}}$ characterizes the PF benchmark. This is expected since a high baseline corresponds to reliable occlusion strategies. However, there is a deeper connection between the occlusion strategy and PF benchmark. This originates from the fact, that a random explanation leads to non-zero values $\overline{\text{R-OMS}}$ for both MIF and LIF. Thus, it is conceptually more meaningful to focus on the relevance gain (RG), as the improvement over the random baseline:

$$\text{MRG}[\phi] = \overline{\text{R-OMS}} - \text{MIF}[\phi] \qquad \text{LRG}[\phi] = \text{LIF}[\phi] - \overline{\text{R-OMS}}. \quad \text{(higher better)} \tag{4}$$

Importantly, MRG and LRG are now directly comparable without changing the final ranking of XAI methods. The theoretical optimal score (area below/above the random baseline) now explicitly depends on the random baseline ($\max(\text{MRG}) = \overline{\text{R-OMS}}$ and $\max(\text{LRG}) = 1 - \overline{\text{R-OMS}}$). In other words, the insightfulness of MIF/LIF benchmarks directly depends on the $\overline{\text{R-OMS}}$ score of the occlusion strategy (left panel in Figure 5).

**Experimental verification** We can also observe this phenomenon empirically by measuring the spread between the performance of different XAI methods. A larger spread (separation of PF curves) indicates a more insightful benchmark. To quantify this spread, we calculate the absolute pairwise differences between the individual PF curves of all XAI methods and average over all pair-wise differences[1]. Then we relate this separation to the $\overline{\text{R-OMS}}$ score of the PF setup. We obtain a positive Pearson correlation for the MIF measure (0.88) and a negative correlation for LIF (-0.92). This shows that the insightfulness of the MIF and LIF measures are conversely dependent on the $\overline{\text{R-OMS}}$ score. For LIF, the reliability and insightfulness of the PF setup are not aligned (left panel in Figure 5). Consequently, a medium score is most beneficial and leads to consistent rankings, as visible in Figure 4.

**Combining most and least relevance gains** Previous studies observed that MIF and LIF can lead to disagreeing rankings (Tomsett et al., 2020; Rong et al., 2022). Our analysis revealed that this disagreement is complementary: depending on the random baseline either MIF or LIF are insightful. This motivates to evenly combine both measures[2]. The relevance gain (Equation (4)) allows to aggregate the most and least

---

[1] The difference cancels the offset in Equation (4). Thus, MIF/LIF and MRG/LRG are interchangeably.
[2] Samek et al. (2016) discuss a similar measure in the appendix. However, it has not been picked up on in following literature.

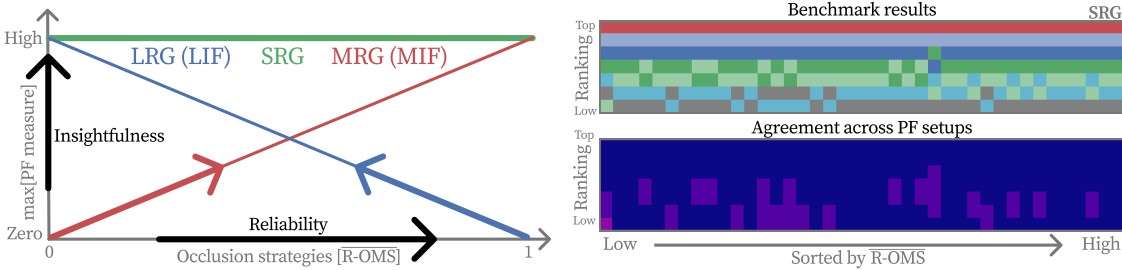

Figure 5: Consistency of SRG measure is independent from the occlusion strategies. Left panel: theoretically achievable improvement over random baseline. Right panel: SRG rankings of XAI methods (legend in Figure 4).

relevant sides of the attribution spectrum into the symmetric relevance gain (SRG)

$$\text{SRG}[\phi] = \text{LRG}[\phi] + \text{MRG}[\phi] = \text{LIF}[\phi] - \text{MIF}[\phi], \tag{5}$$

which corresponds to the area between the PF curves in Figure 1. Trivially, the SRG measure is zero for a random explanation, as the relevance gains MRG and LRG default to zero.

**LRG and MRG measure same notion of faithfulness** Attributions highlight the relevant model reasoning into a single relevance score per feature. This can be regarded as locally approximating the model by an additive function (Lundberg & Lee, 2017). Interestingly, Hama et al. (2023) showed that LRG and MRG converge to the same optimum for such an additive function. Thus, both measures capture the same notion of faithfulness (between attributions and underlying model) and should thus be treated symmetrically. The simplest way of implementing this is the SRG measure in Equation (5). Going beyond this additive (first-order) model structure, requires to estimate interaction effects (Blücher et al., 2022; Bordt & von Luxburg, 2023). Since interactions attribute multiple (joint) relevance scores to a single feature, novel targeted measures are required to capture their faithfulness.

**Theoretical consequence for SRG measure** The theoretical optimal score for both LRG and MRG depends on the $\overline{\text{R-OMS}}$ of the occlusion strategy (as discussed below Equation (4)). This induced the disagreement problems into the final PF rankings (Figure 4). By combining both measures into the SRG measure we can alleviate this undesired dependence. The theoretical optimal SRG score is not bounded by the random baseline and always one (left panel in Figure 5). This leads to *consistent* and *quantitative stable* PF benchmarks (as verified below). In other words, the SRG measures decouples PF and occlusion strategy.

**Consistency of the novel SRG measure** We show SRG rankings for all 40 PF setups in Figure 5 (upper right panel). This leads to only 7 different rankings, which is approximately 3 times less as compared to the one-sided MRG and LRG measures. Moreover, the remaining disagreement (lower right panel) is limited to neighboring and low-ranked XAI methods. In contrast, both MIF and LIF disagree within the top-ranked XAI methods (Table 1). Thus, combining both one-sided measures into the SRG measure leads to consistent rankings across the full spectrum of occlusion strategies.

**Quantitative stability of SRG measure** In addition to this consistency, the SRG measure is also quantitatively stable. This means, that the performance of each XAI method is not affected by the value of the random baseline $\overline{\text{R-OMS}}$. To analyze this, we consider variance for all three measures across PF experiments. Based on the variance of the LRP method across all 40 PF setups, we find that the SRG measure is up to ten times more stable[3]. To visualize this, we show a boxplot of all three measures in the supplementary Figure A3, which validates the above conclusion. In summary, the quantitative stability of the SRG measure allows for aggregating multiple PF setups without losing the underlying ranking of the individual benchmarks. This is not the case for both MRG and LRG.

**Summary** Disagreeing rankings limit the usefulness of PF as a benchmark for XAI methods. The random baseline $\overline{\text{R-OMS}}$ allows to identify insightful PF setups and groups consistent rankings. This re-establishes

---

[3]Variance of the LRP method: $\text{Var}(\text{MRG}) = 0.0110$, $\text{Var}[\text{LRG}] = 0.0128$, $\text{Var}[\text{SRG}] = 0.0005$

Table 6: The SRG measure enables trustworthy PF benchmark of XAI methods, which resolves the disagreement problem from Table 1. Higher is better and random explanations yield a score of zero. Results are consistent for full range of design choices (PF setup: cv2, standard-ResNet50 and $n = 25$) and with a trustworthy MRG benchmark (see supplementary Table A3).

| Occlusion-based methods | | Pixel-wise attributions | |
|---|---|---|---|
| Method | SRG (↑) | Method | SRG (↑) |
| Shapley values (cv2) | 0.47 | LRP | 0.35 |
| Shapley values (Mean) | 0.40 | Saliency (NT) | 0.25 (0.30) |
| PredDiff (cv2) | 0.33 | IG (abs / NT) | 0.23 (0.24 / 0.12) |
| ArchAttribute (cv2) | 0.25 | InputXGradients (abs) | 0.05 (0.19) |

trust in the conventional MIF/LIF measures. Using the full PF information (most + least) leads to the SRG measure. This obviates the disagreement problem, as the SRG measure is decoupled from the random baseline.

# 6 Trustworthy PF benchmarks for XAI methods

Having discussed all the prerequisites for reliable and insightful pixel flipping experiments, we are now ready to perform a final benchmark comparing all major XAI methods. To this end, we show the results for the SRG measure in Table 6. These results align with the MRG measure for a reliable and insightful occlusion strategy, i.e. high $\overline{\text{R-OMS}}$ score, which are presented in the supplementary Table A3. This singles out the diffusion imputer, however at the cost of drastically increased computational costs. In contrast, the SRG measure does not depend on the occlusion strategy and can build on a cheaper strategy (e.g. cv2). We restrict to $n = 25$ superpixels to ensure fully converged Shapley value attributions.

**Occlusion-based vs pixel-wise attributions** The final ranking in Table 6 clearly shows that occlusion-based attributions are significantly more faithful to the model than pixel-wise attributions. This advantage comes at the price of higher computational cost. To emphasize this, note, that LRP only requires a single backward pass. In contrast, the cheapest occlusion-based attribution method, PredDiff, scales linearly with the number of superpixels with a two-orders of magnitude pre-factor (Blücher et al., 2022). All other occlusion-based approaches are significantly more expensive. LRP is considerably more efficient but still achieves faithfulness scores in the same range as PredDiff.

**Matching occlusion strategies for Shapley values and PF** For model-agnostic XAI methods, it is possible to match the occlusion strategy to the PF setup. Thus, a natural question is how much this alignment improves the PF performance. To estimate this effect, we calculate Shapley values using all numerically feasible imputers and report the worst performing imputer in Table 6. We report the results for all possible imputer combinations of the SRG and MIF measure in supplementary Table A2. This analysis confirms that using matching imputers for both attribution and PF benchmark leads to the best performance. However, Shapley values with a mismatching imputer are still superior to the best-performing alternative XAI methods (PredDiff and LRP). This is very reassuring since it verifies that PF benchmarks are not dominated by the choice of occlusion strategy.

**Pixel-wise attributions** We now focus on the differences between the pixel-wise attribution methods. Here, the most surprising observation is that saliency-based explanations are more faithful than the axiomatically grounded IG counterpart. Interestingly, employing a noise tunnel only improves the faithfulness of saliency whereas it is detrimental for IG. Note, that we restrict to the default zero baseline for IG and do not test alternatives (Sturmfels et al., 2020). As saliency builds on the absolute-valued gradients, we also report the absolute-valued IG performance, which is slightly improved (see also supplementary Figure A2). Overall, we suspect that the superpixel average of gradients is sufficient to provide a clear signal for the feature relevance (Kapishnikov et al., 2019; Muzellec et al., 2023). We stress, that this also holds for many small superpixels ($n = 5000$), which were incorporated in the previous set of occlusion strategies (Figure 5).

# 7    Conclusion

This study analyzed the inherent connection between PF benchmarks and the employed occlusion strategies. This connection leads to contradicting rankings and thereby limits the usefulness of PF as a tool to identify faithful XAI methods. We resolve this disagreement problem by disentangling two central ingredients: reliable occlusion strategies and insightful PF setups.

We propose to characterize occlusion strategies based on the R-OMS score. The R-OMS score measures how much information about the original sample is still contained in the occluded samples as perceived by the model. Thereby, we can capture dominant differences between strategies across all relevant design choices. This allows to identify reliable occlusion strategies without prior knowledge about the model architecture or training procedure. Additionally, the R-OMS score indicates insightful PF setups when removing either the most or least influential features first. This groups consistent rankings for both conventional MIF and LIF measures. Importantly, insightfulness and reliability of the occlusion strategy are aligned for the MIF measure, which leads to an overall trustworthy PF setup. Moreover, symmetrically combining both possible feature orderings into the SRG measure, entirely breaks the troublesome connection between the occlusion strategy and the PF benchmark. This circumvents expensive PF setups, as required for trustworthy MIF benchmarks, which require building on a diffusion imputer. The SRG measure consistently evaluates the faithfulness of XAI methods across all design choices. We expect that this insight will improve the comparability of future studies and will thereby foster sustainable progress toward a generally acknowledged understanding of XAI.

**Limitations and future work** This study focused on image classification as the prototypical use case of XAI methods. To explore possible limitations systematically, we first consider other input domains and then move forward to different output domains (tasks). Generally, PF benchmarks require custom imputers for each new input domain. This means that the particular design choices of the occlusion strategy depend on the data domain. For example, it can be challenging to identify appropriate analogues of superpixels for tabular or time series data. Thus, a systematic exploration of plausible occlusion strategies based on the R-OMS score seems to represent a promising direction for future research. Complementary, within the output domain, our proposed methodology is tied to the classification scenario, as we used the remaining class probability to characterize occlusion strategies. This might not be possible for other targets such as multi-class predictions or physical regression targets (Hama et al., 2023). Here, a single output score, with the simple interpretation of lower is less faithful, might not be applicable. However, symmetrizing the PF measure (as done for the SRG measure) should still be beneficial. Going beyond a single score for characterizing imputers could also be interesting even within a classification scenario. Two imputers with similar model outputs can still trigger different intermediate representation, which means that the samples are perceived differently by the model. To capture such effects one can measure the similarity of imputed and reference sample based on intermediate features. Here, other measures such as the Mahalonobis distance (Mahalanobis, 1936) or concepts activations (Kim et al., 2018; Vielhaben et al., 2023) are imaginable. To keep the presentation self-contained, we focused on the simple R-OMS score based on output probabilities, but first exploratory results in this direction can be found in Appendix A.3. In particular, invoking a R-OMS score based on intermediate features also alleviates the restriction to the classification setting.

# Acknowledgments

SB and JV gratefully acknowledge funding from the German Federal Ministry of Education and Research under the grant BIFOLD22B, BIFOLD23B and BIFOLD (01IS18025A, 01IS180371I). The authors thank K.-R. Müller, S. Letzgus, L. Linhardt and S. Salehi for helpful comments on the manuscript.

# A   Additional experiments

## A.1   Details on characterizing occlusion strategies

The model variance is estimated based on the setups provided in Table A1. Each setup corresponds to one column in a single figure of the lower panel (B, C, & D) in Figure 3.

Table A1: Setups for variance of the model design choice in Figure 3.

| # | 1 | 2 | 3 | 4 | 5 |
|---|---|---|---|---|---|
| Imputer (PF) | Train set | cv2 | cv2 | Diffusion | Mean |
| $n$ | 25 | 75 | 75 | 200 | 5000 |
| Shape | Standard | Squares | Semantic | Standard | Standard |

## A.2   Matching imputer for Shapley values and PF assessment

Occlusion-based explanations can match the occlusion strategy of the PF setup. This provides an inherent advantage over XAI methods. To estimate this effect, we compare matching versus non-matching imputer distributions in Table A2. We report both SRG and MIF measure for the standard-ResNet50. Clearly, matching imputer distributions lead to the best results.

Table A2: Ranking Shapley values based on different imputers ($n = 25$, standard-ResNet50) with all possible PF setups. Matching the imputer for PF assessment and attributions is the most superior.

| Imputer (PF) | Mean | Train set | Histogram | cv2 |
|---|---|---|---|---|
| | **SRG** | | | |
| Ranking | **Mean** | **Train set** | Train set $(-1.5)$ | **cv2** |
| attributions | Train set $(1.0)$ | Histogram $(60.9)$ | **Histogram** | Train set $(47.6)$ |
| ($\Delta$ SRG) | Histogram $(10.6)$ | Mean $(65.2)$ | Mean $(6.6)$ | Histogram $(62.1)$ |
| $[10^{-3}]$ | cv2 $(43.5)$ | cv2 $(72.4)$ | cv2 $(35.8)$ | Mean $(70.1)$ |
| | **MRG** | | | |
| Ranking | **Mean** | **Train set** | **Histogram** | Train set $(-0.7)$ |
| attributions | Train set $(1.4)$ | Mean $(10.2)$ | Train set $(5.2)$ | **cv2** |
| ($\Delta$ MIF) | Histogram $(1.4)$ | Histogram $(11.2)$ | Mean $(6.0)$ | Histogram $(10.5)$ |
| $[10^{-3}]$ | cv2 $(11.4)$ | cv2 $(14.7)$ | cv2 $(14.5)$ | Mean $(11.1)$ |

## A.3   Diffusion imputer matches internal strategy

We analyze occlusion strategies for a ViT model in Figure A1. The left panel builds on the R-OMS score. The general behavior is comparable to the timm-ResNet50 in Figure 3 (A). For the ViT model, we can additionally compare to an internal strategy (omitting tokens). Interestingly, the R-OMS score of internal and diffusion strategy are closely aligned. This is unexpected since both occlusion mechanism are conceptually different.

This raises the question whether both strategies are perceived similar by the ViT model. To analyze this, we go beyond the R-OMS score, which quantifies the impact of the imputer through a single number. However, this can lead to identical R-OMS-score even though the imputed samples differ qualitatively. For a more detailed evaluation, we propose to compare feature representations of the original image and occluded samples (Crothers et al., 2023). The right panel in Figure A1 shows that diffusion and internal strategy are also aligned for the intermediate hidden features. This alignment is an interesting argument in favor of the diffusion imputer as a natural replacement strategy.

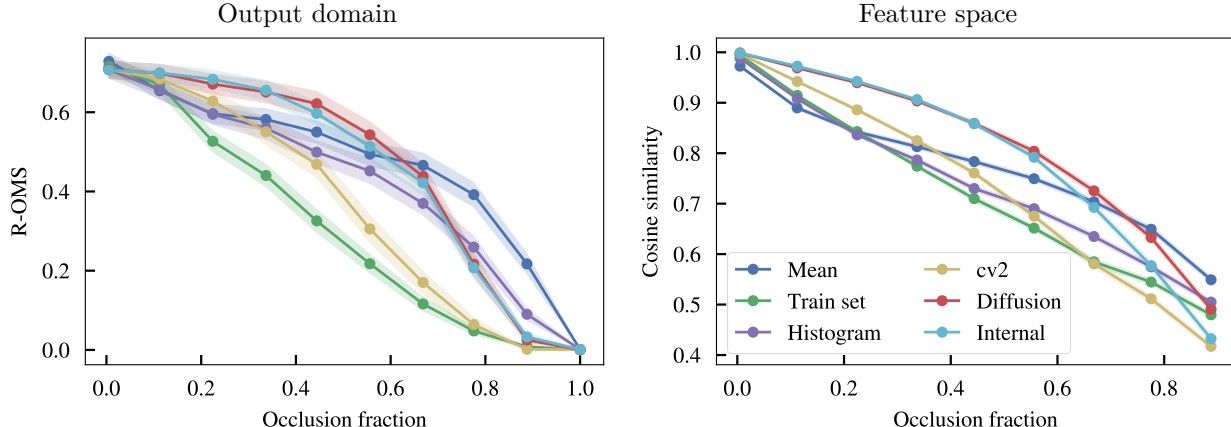

Figure A1: ViT model: Comparing internal strategy (omitting tokens) with external imputations. Left panel: comparison based on R-OMS score. Right panel: similarity in feature space of remaining tokens. The diffusion imputer closely resembles the internal strategy.

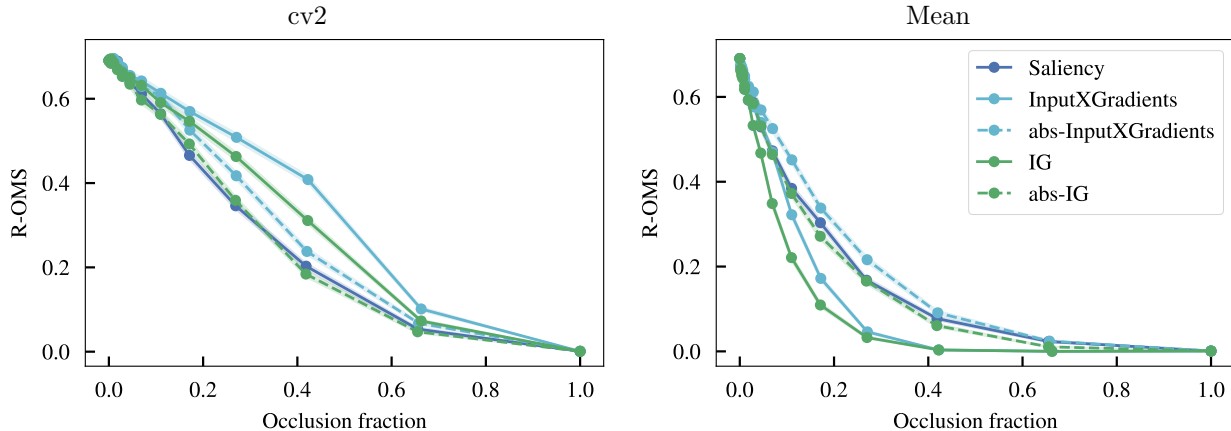

Figure A2: Using the signed or absolute valued gradient attributions has a consistent effect on the performance. ($n = 500$ and standard-ResNet50)

## A.4  Effects of absolute value for gradient-based attributions

Figure A2 provides a more details on the effect of using the absolute valued attributions. Clearly, the performance signed and non-signed attributions are more aligned. In particular, this also holds when IG is more faithful as Saliency (right panel).

## A.5  Quantitative stability of SRG

The SRG measure combines the insightfulness of MIF and LIF. Therefore, it is independent of the random baseline $\overline{\text{R-OMS}}$ and leads to quantitative stable scores for a fixed method across several PF setups. This is shown in Figure A3 by averaging across design choices. Here, we condition the boxplot on the imputer as the dominant design choice. For the LRG and MRG measure, we observe a large spread for the top-ranked methods across the different imputers. This is caused by the strong dependence on the random baseline. As a consequence the original (most frequent) ranking is only visible for insightful imputer, i.e., reliable imputers (cv2, diffusion) for the MRG measure and simple imputers (mean, train set, histogram) for the LRG measure. This is to be contrasted to the SRG measure in the middle pane. The ranking is recognizable

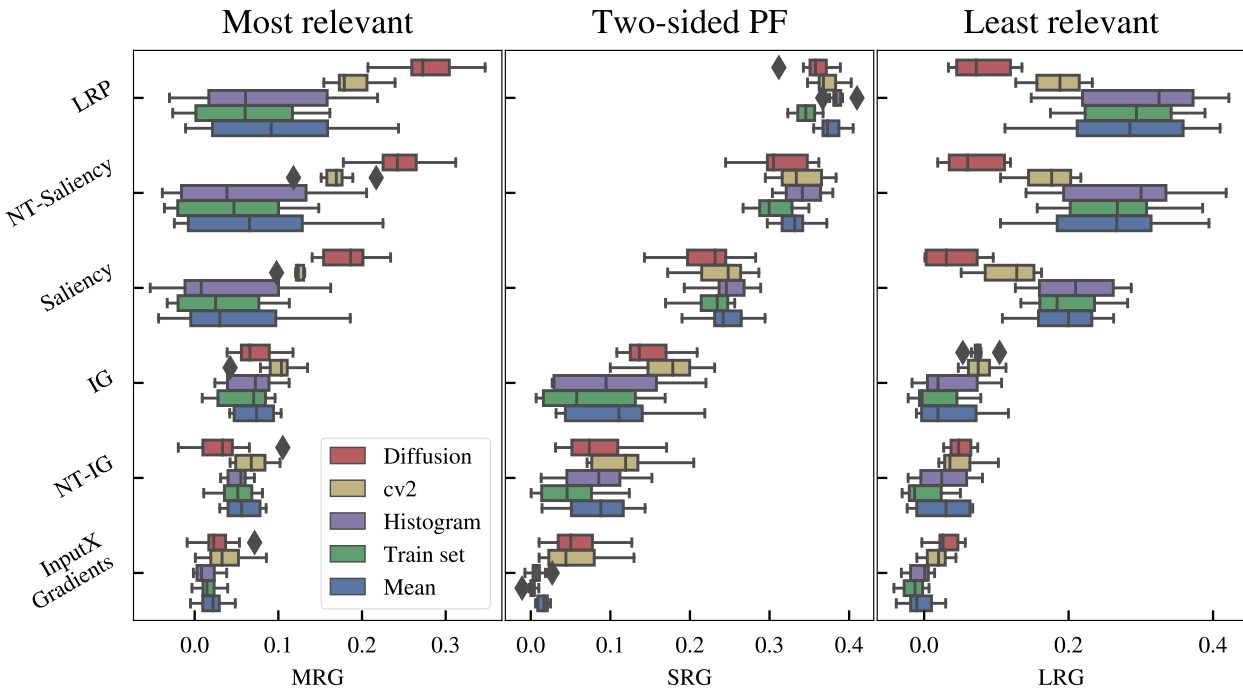

Figure A3: Average (over design choices) performance of XAI methods for the three different PF measures. XAI are sorted according to the most frequent ranking in the y-label. When methods (boxes) are separated horizontally we can infer ranking. The SRG measure is stable and the underlying ranking is still visible. In contrast, the LRG and MRG measure are numerically unstable.

Table A3: The MRG measure with a high $\overline{\text{R-OMS}}$ ensures a trustworthy PF benchmark of XAI methods, which resolves the disagreement problem from Table 1. Higher is better and random explanations yield a score of zero. (PF setup: diffusion, standard-ResNet50 and $n = 25$).

| Occlusion-based methods | | Pixel-wise attributions | |
|---|---|---|---|
| Method | MRG ($\uparrow$) | Method | MRG ($\uparrow$) |
| Shapley values (Train set) | 0.22 | LRP | 0.21 |
| Shapley values (Mean) | 0.20 | Saliency (NT) | 0.15 (0.18) |
| ArchAttribute (Train set) | 0.19 | IG (abs / NT) | 0.10 (0.14 / 0.06) |
| PredDiff (Train set) | 0.18 | InputXGradients | 0.02 |

for all five imputers. This means that the SRG measure is numerically stable with regard to varying design choices. Phrased differently, the SRG measure is independent from the random baseline $\overline{\text{R-OMS}}$.

## A.6 Trustworthy MRG benchmark

The SRG measure allows to perform trustworthy benchmarks independently from the occlusion strategy. However, the MRG measure can still be used if an occlusion strategy with high $\overline{\text{R-OMS}}$ score is invoked. This ensures an insightful measure and a reliable occlusion strategy. Therefore, we present an additional overview benchmark based on the MRG measure in Table A3. Note, that this result relies on the expensive diffusion imputer.

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
