# OpenReview forum: "Decoupling Pixel Flipping and Occlusion Strategy for Consistent XAI Benchmarks"
_TMLR — Accepted by TMLR_

### Review · Reviewer_xdiZ · 2024-03-21

**Summary Of Contributions:**

This paper studies, and claims to finally resolve, "the disagreement problem" for explainable AI methods. The paper makes a slightly odd definition of the disagreement problem, but in the related work this is defined as the issue where multiple AI explanation methods arrive at inconsistent explanations. The paper focuses especially on two "feature removal" methods: occlusion, and pixel flipping. The paper proposes a new scoring system for occlusion methods, evaluating "reference" "out-of-model-scope-ness", and also a score for measuring feature relevance, called "symmetric relevance gain".

**Audience:**

Yes

**Claims And Evidence:**

No

**Requested Changes:**

If I were working on the paper, I would start by rewriting the introduction. I recommend greatly narrowing the scope, simplifying the writing, clarifying the story and the key ideas, and supporting each claim with appropriate references or results. In particular, it is very bad to make a claim and then cite a paper that does not support the claim.

**Strengths And Weaknesses:**

First let me state that I am not an ideal reviewer for this paper. I do not work in explainable AI, so I had to dig through the references to see what is going on.

The paper has a variety of claims that I think are wrong.

The paper says "many competing XAI methods exist, which has sparked criticisms from both theoretical and practical perspectives" (references omitted for brevity). I checked the references here, and they do not seem to support this claim. For example, Doshi-Velez & Kim (2017) propose a taxonomy of evaluation approaches for interpretability. This is very different from criticizing the fact that multiple methods exist. I also checked Rudin (2019), and this paper argues for creating inherently interpretable models instead of searching for explanations post-hoc. Again, this is very different from the stated claim.

The paper goes on to state that the criticism toward competing XAI approaches has led to "the disagreement problem, which summarizes the inconclusive scenario of multiple, contradictory explanations". I checked the paper referenced here, and this is not quite the disagreement problem as defined by that paper. It has nothing to do with summarizing an inconclusive scenario. It actually refers to the issue where explanations obtained by different methods disagree with each other. Also, I could not find support in that paper for the idea that the disagreement problem came from criticism of multiple competing XAI methods.

The paper then claims "this naturally arises" for occlusion-based explanations (without defining what occlusion-based explanation means, or what exactly arises), and states that the problem "extends" to pixel flipping benchmarks (again without defining these). The paper also claims that removing most-influential or least-influential features is "complementary to the ambiguity", which I don't understand.

The largest claim is that this work "resolves the disagreement problem". Given the scope of the problem as defined in Krishna et al. (2022, which introduced this term), and given that the current paper defines the problem in a different (or at least odd and confusing) way, I am extremely skeptical of this huge claim. Even if the errors in definitions are somehow typographical errors and the results should be interpreted as tackling the actual disagreement problem, there is no way that this paper actually resolves the problem since it does not consider the same set of methods and tasks as in Krishna et al.: LIME, Kernel SHAP, Gradient Input, SmoothGrad, Integrated Gradients, and GradCAM. In particular, GradCAM seems critical to include, since that one has broken through to broader use in computer vision.

---

> ### Author Response · Authors · 2024-04-11
> **Improving introduction**
>
> We thank the reviewer for the time and effort reviewing our manuscript. We fully agree with the reviewer that it is instrumental to present evidence for all claims and contributions made within a scientific manuscript. To clarify these, we reiterate our main contributions, highlighting the theoretical and experimental support presented in this manuscript. At this point, we also want to point towards both other reviews, which have greatly appreciated the sound verification of our insights. Along the way, we also outline all changes implemented within the introduction to circumvent future misunderstandings for readers outside the domain of XAI.
>
> **Objective comparison of occlusion strategies** (*Contribution 1*) \
> Occlusion strategies are a mandatory component of many XAI methods (Sec. 2). However, they are subject to multiple design choices (Sec. 3, choice of imputer or number/shape of superpixels). This leaves practitioners and researchers puzzled, on which occlusion strategy is applicable in their scenario in question.  Previously, no objective procedure to resolve this ambiguity existed in the literature. Here, we propose the R-OMS score, which measures the remaining model confidence for the original reference class. In Fig. 3 we show that a large spectrum of occlusion strategies can be characterized by R-OMS score. This provides an objective tool to distinguish different occlusion strategies. Additionally, we show that the R-OMS score captures the relevant differences within disagreeing pixel flipping benchmarks (see Contr. 2).
>
>
> **Disagreement problem of pixel flipping benchmarks** (*Clarification*) \
> In recent years, the XAI community has developed a plethora of different methods [Samek et al., 2021]. Some of which have particular properties and should be used appropriately [Freiesleben & König, 2023]. Other methods can be used interchangeably, such as all saliency methods (see Tab. 6) used in this manuscript. To describe the scenario of multiple (contradicting) explanations the term disagreement problem was dubbed in Krishna et al. (2022). A promising possibility to tackle this problem is to falsify disagreeing explanations via some objective (external) criterion. This can be done via a pixel flipping benchmark, which is a popular method to evaluate the faithfulness of XAI method. However, we show that PF benchmarks themselves are subject to disagreeing rankings (see Tab. 1). As both problems are of the same nature, we extend the terminology disagreement problem to the PF benchmark. To clearly distinguish the two, we now call this the disagreement problem of PF benchmarks throughout the revised manuscript.
>
> **Resolving disagreement problem of PF benchmarks** (*Contribution 2*)\
> The disagreement problem of PF benchmarks states that different setups lead to contradicting rankings of XAI methods. We exemplify this in Tab. 1 and present a detailed analysis in Fig. 4 based on popular 6 saliency methods. In particular, the ranking agreement is correlated with the previously established R-OMS score (contribution 1). In the next step, we propose the SRG measure and show all resulting rankings in Fig. 5. After inspecting the disagreement heatmap (right bottom panel) it is clear that all disagreement between different PF setups disappears. This validates the claim that the SRG measure resolves the disagreement problem of PF benchmarks.
>
> **Rephrasing the introduction**\
> We regret that the reviewer got stuck in the starting lines of our manuscript. Our intention was to set the general scene and give related notable references. These should have served the interested reader as a starting point for a more in-depth journey on the current challenges of the field of XAI. Based on the reviewer’s comments we agree that the selected references did not appropriately match the stated claims. To correct this we have removed unnecessary (too loosely related) references in the revised version of the manuscript. Further, we have improved the description of the disagreement problem, occlusion-based XAI and pixel flipping benchmarks based on the reviewers comments.
>
> We thank the reviewer again for putting in the effort and helping us improve our manuscript. It is of crucial importance for us to ensure a clarified story line for a broad readership.

---

> > ### Comment · Reviewer_xdiZ · 2024-05-03
> > **Thank you**
> >
> > I checked the revisions and I think the paper has improved a lot.
> >
> > I am a little worried that the revisions primarily cover my specific complaints. I hope the authors put the same effort into the areas that I am not equipped to evaluate myself. It doesn't matter very much that the other reviews were overall positive -- the best-case scenario is that more-critical readers will come.

---

### Review · Reviewer_9bnw · 2024-03-25

**Summary Of Contributions:**

1. The study dissects the connection between pixel flipping (PF) benchmarks and the various occlusion strategies employed in XAI. By revealing the inherent link between these two components, the submission provides a pathway to resolving inconsistent rankings and benchmarks.
2. Validation Across 40 Occlusion Strategies. The submission provides empirical validation of its proposed methodologies across 40 different occlusion strategies, demonstrating their effectiveness in resolving the disagreement problem and enhancing the trustworthiness and comparability of PF benchmarks in XAI.

**Audience:**

Yes

**Claims And Evidence:**

Yes

**Requested Changes:**

It is critical to include a comprehensive discussion of potential limitations or assumptions underlying the proposed methodologies.

**Strengths And Weaknesses:**

The study validates its proposed methodologies across a wide range of occlusion strategies, demonstrating their effectiveness in resolving inconsistent rankings and benchmarks. This empirical validation enhances the credibility and applicability of the proposed approaches.

While the submission discusses the strengths of the proposed methodologies, it could benefit from a more thorough exploration of potential limitations or assumptions underlying these approaches. A deeper discussion on the practical constraints or scenarios where the proposed methods might not be suitable would enhance the robustness of the study.

---

> ### Author Response · Authors · 2024-04-11
> **Limitations and practical constraints**
>
> We thank the reviewer for his/her positive assessment of this manuscript.
>
> **Limitations and practical constraints** \
> We agree with the reviewer that it is important to discuss limitations and underlying assumptions of research findings. We regret that this was not clear enough in the previous version of the manuscript. We address this by adding a dedicated paragraph at the end of the conclusion section. In summary, our major limitation is the restriction to image classification within the empirical evaluation of our methodology. We now discuss how parts of the methodology could be transferred to other modalities such as text/speech or different tasks.

---

### Review · Reviewer_FCxG · 2024-04-01

**Summary Of Contributions:**

This paper addresses an important issue with pixel flipping (PF) benchmarks used to evaluate feature attribution methods in explainable AI (XAI). The authors point out that PF rankings of XAI methods are sensitive to the choice of occlusion strategy used to remove features, leading to contradictory rankings. To resolve this disagreement problem, they propose to characterize the reliability of occlusion strategies using a Reference-Out-of-Model-Scope (R-OMS) score that measures how much information about the original sample is retained after occlusion. This enables comparing occlusion strategies objectively. In addition, they combine the most-influential-first (MIF) and least influential-first (LIF) relevance gains into a symmetric relevance gain (SRG) measure. The SRG breaks the dependence on occlusion strategy and leads to consistent XAI method rankings across occlusion setups.

**Audience:**

Yes

**Claims And Evidence:**

Yes

**Requested Changes:**

Please address the weaknesses in the revision.

**Strengths And Weaknesses:**

Strengths:
1. The paper convincingly demonstrates the disagreement problem in PF benchmarks and traces its root cause to the choice of occlusion strategy. This is an important issue that needs to be addressed.
2. The proposed R-OMS score provides a principled way to compare occlusion strategies and identify reliable ones. The experiments systematically analyze the impact of various design choices.
3. The SRG measure is a clever way to decouple PF rankings from the occlusion strategy by combining MIF and LIF. The experiments show SRG provides stable rankings across occlusion setups.
4. The paper conducts extensive experiments, considering 40 occlusion strategies. The final XAI method ranking using SRG on a reliable occlusion strategy is a valuable benchmark for the community.
Weaknesses:
 1. While the SRG measure works well empirically, a deeper theoretical analysis of why symmetrically combining MIF and LIF leads to occlusion-independent rankings would strengthen the paper.
2. The paper focuses on image classification. Discussing the applicability of the approach to other data modalities would be useful, such as language and graphs.
3. The diffusion-based occlusion strategy performs best but has a high computational cost. A more efficient alternative that works nearly as well would be practically valuable.

---

> ### Author Response · Authors · 2024-04-11
> **Theoretical analysis and more details**
>
> We thank the reviewer for the positive feedback and the appreciation of our research.
>
> **Deeper theoretical analysis** \
> We are very grateful for this important remark by the reviewer. To address it we added a new paragraph directly after introducing the SRG measure to concisely present all its theoretical benefits. In summary, both individual MRG and LRG measures target the same notion of faithfulness. Thus, combining both allows to remove the inherent dependence on the occlusion strategy as their weak-spots (less insightful PF setups) are complementary (see left panel in Fig. 5). In turn, this leads to consistent and stable PF benchmarks.
>
> **Different data modalities** \
> We thank the reviewer for this comment. We fully agree that other data domains are an interesting direction of future research. To ensure a concise presentation we restricted this manuscript to image classification. To address this we embedded a discussion of implications of our findings towards other modalities within the new limitation paragraph (see end of conclusions).
>
> **Computational efficiency** \
> We agree that computational efficiency is key for the practical useability of research results. As the reviewer points out SOTA diffusion models are prohibitively expensive and can currently not be used as imputer for occlusion-based attributions methods (e.g. Shapley values). However, improving the sampling speed for example via distillation [1] is a very active area of research. In particular the conventional MIF measure requires reliable imputations (diffusion imputer) to ensure a consistent PF benchmark (supplementary Tab. A3). Interestingly, we show that this can be circumvented based on the SRG measure, which does not depend on the underlying occlusion strategy. Therefore, the much cheaper cv2 imputer can be invoked (see benchmarks results in Tab. 6). In turn this ensures the computational efficiency of the overall PF approach.
>
> [1] Salimans, Tim, and Jonathan Ho. "Progressive distillation for fast sampling of diffusion models." arXiv:2202.00512 (2022).

---

### Author Response · Authors · 2024-04-11
**Overview and revised manuscript**

We appreciate the generally positive comments by the reviewers.  No reviewer addressed any technical or methodological concerns, which would have necessitated additional experiments to validate our contributions. We especially welcome the constructive suggestions made by the reviewers to improve the presentation of our work, which we have fully taken into account in our revised manuscript.

**Reviewer FCxG** strongly agrees with the contribution of our manuscript and emphasizes the significance of our findings for the field of XAI. His/her comments encouraged us to clarify the theoretical underpinning of the SRG measure, which we have done in a dedicated paragraph in the revised manuscript.

**Reviewer 9bnw** gives a similarly positive assessment of our work. In particular, he/she acknowledges the experimental rigor of research and the immediate value for the XAI community. We have incorporated his/her suggestions to clarify the limitations of this study to improve on the robustness of our findings.

**Reviewer xdiZ** We thank the reviewer for his/her time to review our manuscript, since he/she is “not from the field of eXplainable AI (XAI)”. It helped us clarify the story line (remove unnecessary references) within the introduction to ensure accessibility for both novice and expert readers of our manuscript.

We posted point-by-point responses to all reviewers comments below. In the revised manuscript, we have addressed all specific suggestions and calls for clarification. Revisions in the text are highlighted in orange. We hope that the revised manuscript will now be deemed appropriate for final publication in Transactions on Machine Learning Research.

---

### Decision · Action_Editor_s5kb · 2024-06-10

**Recommendation:** Accept as is

**Comment:**

This manuscript tackles the discord surrounding Pixel Flipping (PF) benchmarks in the context of consistent XAI. It introduces a more robust model evaluation metric, the Reference-Out-of-Model-Scope (R-OMS) score, and devises a Symmetric Relevance Gain (SRG) measure to mitigate conflicting rankings resulting from competing PF approaches: most influential first (MIF) or least influential first (LIF). Following detailed feedback, the authors have adeptly revised the manuscript, swaying both the reviewers and AE. The AE finds the work valuable and recommends its publication in TMLR, advocating for its acceptance.

**Audience:**

Yes

**Claims And Evidence:**

Yes